# How Does Nature Connectedness Improve Mental Health in College Students? A Study of Chain Mediating Effects

**DOI:** 10.3390/bs15050654

**Published:** 2025-05-12

**Authors:** Chong Ma, Mei Zhao, Yuqing Zhang

**Affiliations:** 1State Key Laboratory of Cognitive Science and Mental Health, Institute of Psychology, Chinese Academy of Sciences, Beijing 100101, China; machong0924@126.com (C.M.); zhangyq@psych.ac.cn (Y.Z.); 2Department of Psychology, University of Chinese Academy of Sciences, Beijing 100049, China

**Keywords:** nature connectedness, mental health, resilience, meaning in life, sequential mechanism, college students

## Abstract

How does nature connectedness influence college students’ mental health? To further understand the dynamics at play, this present study delves into the chain mediating roles of resilience and meaning in life, from the perspective of the natural environment’s impact on mental health. In April 2024, researchers utilized quantitative research methods to analyze questionnaire data from 703 college students in China, assessing multiple dimensions such as nature connectedness, resilience, meaning in life, and mental health. The results show the following: (1) All pairs of variables showed significant correlations. (2) Resilience is partly mediated by nature connectedness and mental health. (3) Meaning in life is found to have a partial mediating effect, further elucidating this relationship. (4) A chain mediating role is played by resilience and meaning in life in the connection between nature connectedness and mental health. This study expands the research scope of health psychology, has interdisciplinary research significance, and furnishes theoretical support and important guidance, which are essential in improving college students’ mental health.

## 1. Introduction

The mental health of college students has been a continuing focus of psychological research. Data from the American College Health Association in 2020 revealed a high prevalence of mental disorders among U.S. college students, with 20-45% meeting criteria for at least one condition, primarily depression and anxiety ([2]). China Youth Daily, China Youth School Media, and Dr. Clove jointly released the “2020 Survey Report on the Health of Chinese College Students”, which covered 12,117 college students from more than 40 colleges and universities in China. It comprehensively analyzed college students’ physical and mental health, with only nearly 30% of college students who were assessed that their mental health was in optimal mental health status ([11]). The report in the 2022 edition of the “Blue Book of Mental Health” points out that the detection rate of depression risk in adults is 10.6%, the detection rate of anxiety risk is 15.8%, the depression risk shows a tendency of aging, and the detection rate of depression risk in the age group of college students is 24.1%, which is significantly higher than that in the other age groups ([33]). Facing the severe trend of mental health issues among college students, in addition to traditional psychological intervention, it is of great significance to explore the mechanism by which the natural environment functions as a potential protective factor.

As discussed in this study, “nature” refers to areas comprising non-human species, geomorphic landscapes, and host elements of living systems ([5]). Historically, humans spent most of their time coexisting with nature, relying on the natural environment for survival and reproduction, where exposure to nature was an unintentional part of daily life ([42]). However, with the progress of the agricultural and industrial revolutions and the gradual urbanization of society, a lifestyle marked by alienation from nature emerged ([15]). Technological advancements have significantly altered human–environment interactions, reducing direct engagement with natural systems ([90]). Longitudinal research suggests inverse correlations between nature contact frequency and mental health disorder incidence ([55]). Nature contact—specifically defined in this context as physical-level environmental interactions—encompasses human engagement with natural environments and non-human species ([68]). Studies demonstrate its significant benefits for cognition, emotion, and behavior ([38]). While exposure to natural environments demonstrates short-term benefits, sustained psychological well-being requires measurable nature connectedness ([7]). Therefore, there are important research implications for exploring the impact of nature connectedness on mental health. While distinct from nature contact, nature connectedness also correlates with it. Evidence indicates that both exposure to nature and psychological connection enhance individual health ([111]) and pro-social behavior ([112]), with nature connectedness playing a mediating role between natural contact and individual health ([60]).

The concept of nature connectedness originates from the biophilia hypothesis proposed by American biologist Wilson. This hypothesis posits that humans have inhabited specific natural environments throughout evolutionary history. This proximity to certain plants and animals reflects an instinctive desire that attracts humans towards nature and fosters their dependence on it ([97]). Deep nature connectedness is essentially rooted in an individual’s intrinsic perception of nature’s inclusiveness ([75]), with its core mechanism reflected in the cognitive reconstruction of the “symbiotic relationship between humans and nature” ([65]). This connection is dynamically constructed through the individual’s embodied experience within the natural environment and eventually crystallizes into a stable affective–cognitive bond ([59]). In this study, nature connectedness is interpreted from an emotional perspective, indicating a sense of connection and oneness with nature ([59]). Research has shown that nature connectedness exerts numerous positive effects on human psychology and behavior ([70]), particularly on mental health. Studies indicate that nature connectedness promotes subjective well-being and life satisfaction ([59]) and increases pro-environmental behaviors ([20]). The higher an individual’s level of nature connectedness, the greater their participation in outdoor activities and the positive correlation with the personality trait dimensions of openness to experience, agreeableness, and conscientiousness ([65]).

As a well-established construct in environmental psychology ([20]), nature connectedness has been validated for promoting mental health through affective-cognitive bonds ([70]). With increasing urbanization and associated reductions in nature exposure ([53]), exploring the psychological mechanisms of human–nature relationships has become an important direction for interdisciplinary research ([106]; [51]). While nature connectedness may be one of the ways to improve the mental health of college students, few studies have explored the relationship and the internal mechanism of action for the Chinese college student population. This study employed a cross-sectional design in Chinese college students to examine nature connectedness on mental health and the potential pathways of their influence, to provide references for how nature connectedness can contribute to the prevention and intervention of college students’ mental health.

### 1.1. The Correlation Between Nature Connectedness and the College Students’ Mental Health

Nature connectedness is essential for individual mental health. Individuals with higher levels of nature connectedness are more interested in elements in natural environments and show a stronger willingness to engage with nature and participate in outdoor activities ([28]). By enhancing self-regulation ([80]), such engagement promotes mental health. Noticing natural elements and being mindful of sensory experiences, such as visual and auditory stimuli in nature, can further strengthen nature connectedness ([72]). The attention restoration theory offers a theoretical framework for understanding how natural environments benefit mental health. The theory posits that, unlike artificial environments that require “focused attention”, natural environments provide a unique form of engagement known as “soft fascination”. This particular mode allows people in natural environments to effortlessly notice features around them, such as flowing streams, birdsong, and swaying trees. This form of attention can aid in the restoration of attention and mental energy, thereby promoting positive emotional responses ([39]). Empirical research has demonstrated that natural environments can help people recover quickly from stressful situations. The inherent beauty of natural environments stimulates positive emotional states and mitigates physiological markers of stress, such as lowering blood pressure and boosting the immune system ([88]). Concurrently, interaction with natural environments has been shown to relieve stress, anxiety, eliminate fatigue, and enhance subjective vitality, as well as positively affect the immune system, thereby reducing depression ([36]; [76]; [101]).

College students represent a particularly active demographic in terms of nature contact. However, prior studies have predominantly focused on the effects of nature connection on pro-environmental behaviors and subjective well-being ([107]; [48]), with limited focus on its comprehensive effects on key indicators of mental health, such as depression, anxiety, and stress. To extend existing evidence, the present study proposes the following hypothesis: among college students, the degree of nature connectedness will serve as a significant negative predictor of their mental health, such as depression, anxiety, and stress.

### 1.2. The Meditating Role of Resilience

Resilience represents a dynamic psychological trait, enabling individuals to mitigate the impact of adverse circumstances ([13]). Among youth, resilience development is influenced by problem-solving skills, emotional regulation, self-efficacy, having social support networks, and so on ([57]). Protective factors that foster resilience function as buffers against adversity ([95]). There are three levels of protective factors ([54]). Individual-level protective factors refer to personal attributes that contribute to perceiving, coping, and responding to adversity ([74]). Family-level protective factors involve social support from family and friends. Community-level protective factors include social and physical environments, with the natural environment recognized as a potential community-level protective factor ([58]). According to stress reduction theory, the restorative functions of the natural environment may contribute to the development of resilience through a dual pathway. Firstly, the immediate stress reduction pathway: nature exposure, such as observing vegetation or listening to birdsong, directly reduces an individual’s physiological stress responses (e.g., cortisol levels) and cognitive fatigue, providing immediate emotional resources for resilience by generating a positive emotional state ([88]). Secondly, there is a sustained reinforcement pathway of nature connectedness. An intervention study showed that individuals’ emotional connection to nature is strengthened when they continue to experience nature exposure through daily active attention to the “good things” in nature ([72]). This connection is manifested in increased sensitivity to natural elements and increased willingness to actively engage with nature ([28]). Individuals with greater nature connectedness utilize natural environments for emotional regulation to facilitate quicker recovery from stress ([88]). At the same time, nature experiences promote an adaptive interpretation of life’s challenges ([87]), further enhancing resilience. Research indicates that nature connectedness enhances individuals’ perceived control and security, promoting adaptive and transformative capacities essential for managing crises and confronting future challenges ([4]). A study involving graduate students who spent five days and four nights in a forest setting showed that being in the natural environment reduced problem-focused stress without reliance on emotion-focused coping strategies. The forest environment enhanced coping efficacy and resilience, enabling participants to reassess situations systematically and address problems more effectively ([83]). Based on the above findings, it can be hypothesized that the stronger the sense of nature connectedness of college students, the higher the level of resilience.

Resilience exerts a profound influence on mental health. Based on regulatory focus theory, highly resilient individuals exhibit enhanced cognitive reappraisal abilities when exposed to adverse events ([6]). Their adoption of an objective and rational perspective facilitates effective situational responses. This process reduces perceived stress and diminishes the intensity of negative emotional experiences ([47]). Individuals with lower levels of resilience show reduced capacity to cope with stressful events to maintain or restore optimal mental health ([37]). Studies indicate that resilience serves as a key protective factor against depression, significantly predicting positive emotions and lower levels of negative emotions in college students. Furthermore, increased resilience correlates with enhanced recovery from negative emotional states ([31]). Collectively, these results indicate a strong link between resilience and mental health.

Considering the preceding analysis of nature connectedness, resilience, and mental health, this study proposes that resilience mediates the relationship between nature connectedness and the mental health of college students.

### 1.3. The Role of Meaning in Life as a Mediator

Throughout history, the quest to understand the meaning of life has been a central concern for humanity. In this study, meaning in life refers to pursuing and striving towards personal goals and missions ([19]). It consists of two dimensions: the cognitive dimension indicates the presence of meaning, which involves individuals assigning meaning to their lives, being clear about their goals and missions in life, and focusing on outcomes; the motivational dimension entails searching for meaning, which describes the process of obtaining significance through the efforts that people invest in constructing a sense of meaning and the purpose in life, emphasizing the ongoing nature of meaning seeking ([78]). Existential psychology posits that the search for meaning is inherently relational ([25]), suggesting that meaning in life is developed within a relational context. The biophilia hypothesis proposes that humans possess an innate tendency to form connections with nature ([97]). The need for relationships has traditionally been considered primarily in the context of human-to-human interactions. Beyond human-to-human bonds, interactions with other living (or nonliving) entities in nature can also satisfy relational needs ([67]). Empirical research indicates that, during periods of social isolation, individuals demonstrated an increased desire to connect with nature, which can, in turn, elevate their level of nature connectedness and buffer the negative effects of inadequate interpersonal connections ([69]). Research has found that human existential fulfillment is intertwined with a relationship with nature ([42]), which can help explore the meaning of life. Empirical studies have indicated that the key components of meaning in life, including psychological self-transcendence, adaptability, rational planning, a sense of being alive, and connectedness, are often derived from experiences in nature ([66]). Additionally, it has been demonstrated that nature connectedness positively predicts college students’ meaning in life; the stronger the nature connectedness, the more they perceive life as meaningful ([91]). Based on the results of the above studies, it is clear that a strong relationship exists between nature connectedness and college students’ meaning in life.

Meaning in life serves as a crucial indicator of the level of mental health. Recognition of one’s value and purpose drives individuals towards self-actualization ([46]), which influences psychological states and behavioral patterns ([105]). Research has indicated that college students with a sense of meaning in life are psychologically healthier ([24]), and they can experience a range of positive emotions, such as self-esteem ([82]), subjective well-being ([63]), life satisfaction, and enhanced enjoyment of life ([26]). Conversely, the absence of meaning in life adversely affects mental health, leading to feelings of emptiness and boredom and exhibiting phenomena such as hollow syndrome and lying flat ([50]). These conditions are associated with psychological disorders like depression, anxiety, and, in extreme cases, suicide ([8]; [113]; [109]). Studies have revealed that those who ascribe a higher level of meaning in life are characterized by possessing more psychological energy and are more willing to take responsibility for their lives. In terms of adversity and psychological distress, these individuals exhibit lower tendencies toward suicidal ideation ([99]) and reduced depressive symptoms and stress ([103]). These findings collectively support a robust association between meaning in life and mental health.

Based on the above discussion, this study hypothesizes that meaning in life mediates the relationship between nature connectedness and college students’ mental health.

### 1.4. The Chained Mediating Effect of Resilience and Meaning in Life

While both constructs demonstrate independent mediating effects, resilience and meaning in life may also serve as sequential mediators between nature connectedness and mental health. Resilience and meaning in life are closely interrelated, with resilience positively predicting meaning in life ([89]). Research has shown that resilience is strongly associated with positive coping styles ([35]). Utilizing positive coping styles in challenging situations aids in developing a coherent belief system, which supports goal attainment. Achieving goals and accumulating experiences related to overcoming challenges contribute to a sense of accomplishment and fulfillment, thus enhancing meaning in life ([30]). Higher levels of resilience among college students correlate with increased optimism and perseverance. Such individuals are better equipped to manage stress, take initiative to overcome challenges, and gain self-growth experience through adversity. The journey helps them appreciate life’s value and significance ([108]). Resilience and meaning in life are significant predictors of post-traumatic growth, and resilience serves as a resource that enables individuals to discover meaning in life and create a better future during challenging periods ([45]).

Consequently, this research further hypothesizes that resilience and meaning in life act as a chain mediator linking nature connectedness to mental health.

### 1.5. The Current Study

Empirical evidence indicates that nature connectedness positively influences college students’ mental health. Furthermore, significant interrelationships exist among nature connectedness, resilience, and meaning in life, with each construct demonstrating independent predictive validity for mental health. Nevertheless, the underlying mechanisms governing these relationships remain poorly elucidated in the extant literature. This research aims to probe the relationship between college students’ nature connectedness and mental health while also exploring the independent and chain mediating functions of resilience and meaning in life (Figure 1). In light of the literature review, four hypotheses have been put forward.

**Hypothesis 1.** 
*Nature connectedness significantly and negatively impacts the prediction of college students’ mental health issues.*


**Hypothesis 2.** 
*Resilience serves as a mediator between nature connectedness and mental health.*


**Hypothesis 3.** 
*Meaning in life mediates the relationship between nature connectedness and mental health.*


**Hypothesis 4.** 
*Resilience and meaning in life mediate the chain between nature connectedness and mental health.*


## 2. Methods

### 2.1. Procedure

Ethical approval for this study was granted by the Ethics Committee of the Institute of Psychology, Chinese Academy of Sciences (Approval No: H23128; Date: 10 November 2023). This study employed a cross-sectional design. Before data collection, all questionnaire participants were informed about this study’s purpose, procedures, voluntary participation, and response confidentiality.

An online survey was administered in April 2024 targeting college students enrolled at seven universities in Shandong, Tianjin, Guangdong, Hebei, Xinjiang Uygur Autonomous Region, Jiangsu, and Hunan, China. Random selection of classes was performed within each university, and cluster sampling was applied to every class. Participants in each class were invited to complete the questionnaire survey on the Wenjuanxing platform.

### 2.2. Participants

In this study, 1085 questionnaires were collected, of which 703 were deemed valid after applying predefined exclusion criteria, with a valid response rate of 64.79%. The sample consisted of 375 (53.34%) male students and 328 (46.66%) female students. Additional demographic details are presented in Table 1.

The inclusion criteria are as follows: (1) undergraduate students from freshman to senior years; (2) graduate students in their first to third years; (3) participants were informed about this study and agreed to take part voluntarily, providing their consent.

The exclusion criteria consisted of the following: (1) response time shorter than 100 s or longer than 30 min; (2) failed to pass the validity check in the Meaning in Life Questionnaire, calculated as the lie-detection index score = (7th item + 9th item)/2 − reverse-scored second item, and, when the absolute value of the lie-detection index score is greater than or equal to 3, the data are judged as lying or careless answer and are deleted in this study.

### 2.3. Measures

In this study, all the scales were borrowed and adopted from the well-established scales developed by previous scholars for measurement, except for the general questionnaire prepared by the researchers.

#### 2.3.1. General Investigation

This was prepared by the researchers, including gender, grade, home locality, student leader, and family (whether the only child).

#### 2.3.2. Connectedness to Nature Scale (CNS)

In this study, CNS was used to measure the nature connectedness variable. Mayer and Frantz developed the scale in 2004 ([59]), and the Chinese version was revised by Li Na and colleagues ([49]). Good reliability and validity have been observed for this measure within the population of Chinese college students. The scale consists of 14 items, and the responses are measured on a 5-point Likert scale, with 1 signifying “strongly disagree” and 5 denoting “strongly agree”, with items 4, 12, and 14 reversely scored. As the scores increase, the level of connectedness to nature also rises. The Cronbach’s α reliability coefficient for the scale used in this study was reported as 0.83.

#### 2.3.3. The Brief Version of the Connor–Davidson Resilience Scale (CD-RISC-10)

In this study, CD-RISC-10 was utilized to measure the resilience variable. The scale was developed by Connor and colleagues ([13]), and Chen Wei and his colleagues revised the Chinese version. When used among Chinese college students, it demonstrated good applicability ([9]). The scale, which consists of 10 items, gauges responses on a 5-point Likert scale (0 = “never”, 4 = “true always”). An increase in scores indicates stronger resilience. For the scale in this research, the calculated Cronbach’s α reliability coefficient was 0.92.

#### 2.3.4. The Meaning in Life Questionnaire—Chinese Version (MLQ-C)

The MLQ-C used in this research was used to measure the meaning in life variable. This scale, adapted by Chinese scholars Wang Xinqiang and colleagues, demonstrates high reliability and validity among Chinese college students ([94]). Comprising 10 items, the scale was rated on a 7-point Likert scale, with 1 signifying “not at all consistent” and 7 corresponding to “completely consistent”. Item 2 is reversely scored. It includes two dimensions: the presence of meaning and the search for meaning. Higher scores indicate greater levels of perceived meaning in life. In this study, the internal consistency coefficients of the presence of meaning and search for meaning were 0.86 and 0.87, respectively. Cronbach’s α coefficient of the overall scale achieved a Cronbach’s α of 0.87.

#### 2.3.5. Depression, Anxiety, and Stress Scale (DASS-21)

Mental health in this study was measured by depression, anxiety, and stress levels using the DASS-21 scale. The scale was initially developed by Lovibond and colleagues ([14]). This study uses the Simplified Chinese version of the DASS-21, adapted by Xi Xu and colleagues, which was employed to evaluate the mental health issues among college students and assess their mental health status. This scale has manifested strong applicability and validity among Chinese college students ([23]). This scale, which is made up of 21 items, covers three dimensions: depression, anxiety, and stress. Responses were recorded on a 4-point Likert scale where 0 represents “not met” and 3 represents “always met”. Higher scores indicate greater severity of mental health challenges and decreased mental health quality. In the current study, the internal consistency coefficients for the depression, anxiety, and stress subscales were 0.90, 0.88, and 0.89, respectively. A Cronbach’s α coefficient of 0.96 was achieved for the overall scale.

### 2.4. Data Analysis

Data analysis for this study was conducted using SPSS 26.0 and the PROCESS macro. Initially, a common method bias test was conducted on the data utilizing Harman’s single-factor approach. Descriptive statistics and Pearson correlation coefficients were calculated for variables including nature connectedness, resilience, meaning in life, and mental health. Subsequently, to evaluate the chained mediating effect, Model 6 of Hayes’ PROCESS macro was utilized with bootstrap sample size of 5000. A significant mediating effect was inferred if the 95% confidence interval of the path coefficients did not contain zero. In the data analysis, statistical significance was determined using a two-sided test, with results considered significant at *p* < 0.05.

## 3. Results

### 3.1. The Control Process for Common Method Deviation

Since the data collection in this study utilized self-report measures, common method bias was a potential concern. Harman’s single-factor test was applied as a diagnostic tool for common method bias. In total, 10 factors were extracted without rotation, each demonstrating an eigenvalue exceeding 1. The first factor accounted for 30.19% of the total variance (<40%). Consequently, no significant common method bias was detected in this study ([84]).

### 3.2. Descriptive Statistical Data and Correlation Analysis

Descriptive statistical and Pearson correlation coefficients are summarized in Table 2. Correlation analysis revealed that nature connectedness was positively correlated with resilience and meaning in life and negatively correlated with mental health issues. Furthermore, there is a notable positive correlation between resilience and meaning in life, while resilience shows a clear negative correlation with mental health issues. Finally, a significant inverse relationship was observed between meaning in life and mental health issues.

### 3.3. The Significance Examination of the Mediating Effect

The mediating effect was tested using Model 6 of the PROCESS plugin, with all variables standardized. Covariates included gender, grade, home locality, student leader, and family variables.

Regression analysis (Table 3 and Figure 2) indicated that nature connectedness had a significant negative effect on resilience (*β* = 0.41, *p* < 0.001) and meaning in life (*β* = 0.17, *p* < 0.001) and a significant positive effect on mental health issues (*β* = −0.08, *p* < 0.001). Additionally, resilience had a significant positive predicted meaning in life (*β* = 0.51, *p* < 0.001), and meaning in life exerts a significant negative influence on mental health issues (*β* = −0.19, *p* < 0.001).

According to the hypothesis, a chain mediation model was constructed with mental health issues as the outcome, using nature connectedness as the independent variable, with resilience and meaning in life functioning as consecutive mediating variables. Mediation analysis results (Table 4) indicate that resilience and meaning in life serve as significant mediators in the relationship between nature connectedness and mental health issues among college students. The mediating effect is −0.04, representing 15.38% of the total effect of nature connectedness on mental health issues. There are three indirect pathways: (1) Nature connectedness → resilience → mental health issues. The indirect effect is −0.11, and the 95% CI is [−0.15, −0.07], which does not contain 0. (2) Nature connectedness → meaning in life → mental health issues. The indirect effect is −0.03, and the 95% CI is [−0.06, −0.01], which does not contain 0. (3) Nature connectedness → resilience → meaning in life → mental health issues. The indirect effect is −0.04, and the 95% CI is [−0.06, −0.02], which does not contain 0. Path (1) exhibited the strongest mediating effect among the identified pathways.

## 4. Discussion

In this study, we used a questionnaire to survey 703 Chinese college students. This finding highlights that nature connectedness positively predicts college students’ mental health. Furthermore, resilience and meaning in life play an intermediary role in a chain mediation model connecting nature connectedness and mental health. The findings in this study support the four hypotheses proposed. These findings enrich the research on factors influencing college students’ mental health and offer a solid foundation and support for future research.

### 4.1. How Nature Connectedness Directly Influences the Mental Health of College Students

This finding indicates that the level of nature connectedness is crucial to college students’ mental health, and research hypothesis 1 was verified. This is similar to existing research ([64]). This study introduced nature connectedness as an environmental psychology variable into the realm of health psychology, provided a richer understanding by enhancing the existing literature on the impact of natural environments on mental health, and extensively examined the nuanced connection between nature connectedness and mental health among college students, which has crucial interdisciplinary significance.

The stronger an individual’s sense of connectedness, the more pronounced the psychological benefits obtained from the natural environment ([62]). The body, as a medium connecting individuals to the environment ([18]), plays a pivotal role in emotion construction ([86]), with the cognitive–emotional state of an individual significantly context-dependent ([104]). Research indicates that green spaces, such as parks, woodlands, etc., positively impact mental health ([96]). According to attention restoration theory, natural elements like the sound of streams and birdsong can assist in restoring cognitive resources, promoting relaxation, and relieving psychological stress ([40]). The present study further found that college students’ level of nature connectedness was closely related to their mental health status. Establishing a connection with nature promotes college students’ physical and mental health through multiple mechanisms. At the physiological level, nature exposure improves cardiovascular function ([88]), regulates brain activity, and optimizes endocrine activity and immune system responses ([76]). Consequently, it significantly enhances individual vitality, creative thinking, subjective well-being ([73]; [85]; [17]), and other positive psychological factors at the psychological level, which promote physical and mental health. These findings suggest that nature connectedness positively influences the mental health of college students.

### 4.2. How Resilience and Meaning in Life Act as Mediators

The findings indicated that nature connectedness directly accounts for 30.77% of the variance in mental health. Moreover, the mediating effects through resilience and meaning in life collectively explained 53.85% of the remaining variance, suggesting the importance of intrinsic mechanisms. This research identifies that nature connectedness affects college students’ mental health through three indirect paths.

First, resilience partially mediates the effect of nature connectedness on the mental health of college students, validating Hypothesis 2. This result echoes the previous research indicating that nature connectedness positively affects resilience ([22]) and that resilience positively predicts college students’ mental health ([71]). By incorporating all three variables into this study, it was uncovered that resilience plays a mediating role between nature connectedness and mental health. The effect size of this pathway is the largest among the total indirect effects, accounting for 42.31%, indicating that resilience exerts a key role in linking nature connectedness to mental health, a discovery not identified in previous studies. Natural environments act as protective factors of resilience ([58]). And creating a connection with nature can help resilience levels gain traction. Stress reduction theory posits that nature exerts a restorative and stress-reducing effect, resulting in reduced physiological activity and sustained attention, thereby fostering positive emotional states ([1]). Moreover, positive emotional experiences expand an individual’s momentary thought–action repertoire, fostering enduring personal resources ([21]). Specifically, when college students have positive emotional experiences, their emotional regulation self-efficacy may increase, significantly enhancing their resilience ([110]). The present study also revealed that resilience contributes to mental health by effectively coping with stress, anxiety, and depression. Studies have shown that individuals with high resilience can alter their perception of distress, confront difficulties optimistically, and reduce susceptibility to depression ([12]). In the face of negative life events, resilience assists college students in adjusting their mindset, maintaining emotional stability, and buffering the anxiety response, thus reducing the duration of negative emotions ([52]). According to regulatory focus theory, resilience supports college students to sustain good mental health by promoting regulatory orientation, enhancing sensitivity to positive goals ([16]), and accelerating cognitive switching in response to negative emotions ([81]).

Secondly, this study also identified that meaning in life serves as a partial mediator in connecting nature connectedness to college students’ mental health, supporting Hypothesis 3. These findings align with previous research that nature connectedness positively predicts college students’ meaning in life ([10]) and that meaning in life contributes to improved mental health ([29]). Engaging with nature facilitates the ecological expansion of college students’ self-concept and provides a new avenue for forming connections with the broader world ([92]). When college students’ interpersonal needs remain unmet, compensatory adaptive mechanisms are activated ([100]), which prompts individuals to transform natural entities, such as landscapes, plants, and animals, into alternative sources of social support. Empirical studies suggest that this form of ecological compensation can effectively buffer against psychological damage caused by social exclusion ([102]). Furthermore, facing interpersonal challenges increases college students’ tendency to seek connections with nature, potentially increasing their level of nature connectedness ([69]). Establishing an intimate connection with nature may aid college students in reevaluating the value and meaning of life, enhancing their understanding of life events, and facilitating insights into existential truths ([32]). Research indicates that humans’ innate proximity to nature promotes emotional and psychological development and that a relationship with nature facilitates exploration of the meaning of life ([42]). The mental health of college students is closely related to their meaning in life ([78]). Based on meaning management theory, the pursuit of meaning in life is a human need and key to psychological health and well-being ([98]). Meaning in life involves the understanding and acceptance of oneself ([79]), which can enhance college students’ focus on personal strengths, enhance self-esteem, reduce negative emotions like depression, and foster optimism about the future ([61]). Additionally, research has shown that meaning in life yields beneficial effects on psychological functioning and physical and mental health ([77]), including reductions in anxiety, depression, and suicidal ideation ([27]). In this study, the mediating effect of meaning in life between nature connectedness and mental health accounted for a relatively small proportion (11.54%) of the total indirect effect, which may be attributed to the sequential mediation process where nature connectedness on mental health first enhances resilience and subsequently influences meaning in life.

Finally, based on the mediation effect observed, this study further identified that resilience and meaning in life serve as sequential mediators between nature connectedness and college students’ mental health, and the core theoretical presupposition of the present study, thereby confirming Hypothesis 4. Unlike previous studies focusing on single-path mechanisms, this study reveals the progressive transmission mechanism of resilience → meaning in life. Meaning in life is inherently intertwined with individual experiences, encompassing both past achievements and current life encounters ([93]). College students are at a critical stage of meaning systems and are reportedly 37% more sensitive to achievement events than adults ([3]). Research indicates that increased positive life experiences and achievements enhance college students’ perception of their lives as significant and meaningful ([56]). This perception not only enhances their awareness of life’s value—understanding that their existence can influence the world—but also strengthens their meaning in life ([43]). College students with a strong sense of meaning are more motivated to utilize their initiative to solve problems, accumulate growth experiences, and pursue their personal goals ([108]). Moreover, research has demonstrated that college students with elevated resilience levels are more likely to exhibit positive cognitive attitudes and adopt effective coping strategies when facing challenges ([41]). This resilience facilitates a sense of self-worth in social interactions and during dilemmas, contributing to gaining meaning in life among college students ([44]). However, unresolved tensions arising from dilemmas may lead to a distorted or diminished sense of meaning in life ([34]).

In summary, this study investigates the impact of nature connectedness on college students’ mental health and the chain mediating role of resilience and meaning in life, highlighting its significant practical significance. The research on the effect of nature connectedness on college students’ mental health has been enriched, providing a theoretical foundation for prevention and intervention strategies to enhance their mental health through increased nature connectedness.

## 5. Limitations

However, this research is subject to several limitations that warrant careful consideration. First, the questionnaire method adopted in this study is a preliminary cross-sectional exploration. In future research, longitudinal designs or experimental studies could be utilized, and the causal relationships among variables could be further validated using crossover designs. Second, the sample size used in this study was relatively limited. Future research can expand the sample size for investigation. Third, the sample comprises college students from seven universities, providing a broad range. Considering factors such as geographical environment (e.g., northern vs. southern regions, coastal vs. inland areas) and cultural background (e.g., ethnic minority regions vs. economically developed areas), these factors may introduce significant variability in research outcomes. Future research can explore specific groups within a certain region. Fourth, the present study did not distinguish between the relative impacts of duration and frequency of nature contact on the strength of nature connectedness, which could be further investigated in future studies.

## 6. Conclusions and Future Implications

This study investigated the correlation between nature connectedness and the mental health of college students while simultaneously examining the pivotal roles of resilience and meaning in life within this correlation. This study’s findings indicate the following: First, nature connectedness exhibits a positive association with resilience and meaning in life while demonstrating a notable negative correlation with mental health issues; second, resilience and meaning in life function as critical mediators, elucidating the pathway through which nature connectedness affects mental health issues; and, thirdly, nature connectedness influences resilience, which in turn affects the meaning in life, and ultimately impacts mental health. From the perspective of how natural settings influence human mental health, this paper verifies that nature connectedness can directly impact the mental health of college students, with a particular focus on elucidating the mechanisms by which resilience and meaning in life contribute to mental health. This study broadens the scope of health psychology by enriching theoretical investigations into the mechanisms and factors through which the natural environment impacts human mental health while also offering practical insights relevant to interdisciplinary research. This study provides practical suggestions for mental health interventions. First, to enhance nature connectedness, educational institutions may implement nature education programs, organize outdoor activities (e.g., travel, park sports), and utilize indoor activities like viewing nature-themed media and engaging in nature-themed mindfulness training, thereby improving mental health. Second, educational institutions should pay attention to the cultivation of resilience in college students. Educators could carry out frustration education and group training, which can help college students develop effective responses to challenges and adversity. Finally, efforts should be made to strengthen college students’ sense of meaning in life. Schools can adopt approaches such as life education or group counseling to cultivate college students’ strong willpower, improve their pursuit of value and meaning, and promote the enhancement of mental health levels.

## Figures and Tables

**Figure 1 behavsci-15-00654-f001:**
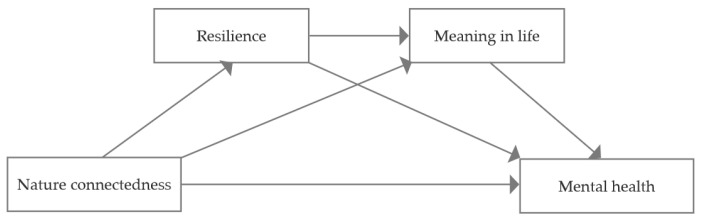
Framework of our research model.

**Figure 2 behavsci-15-00654-f002:**
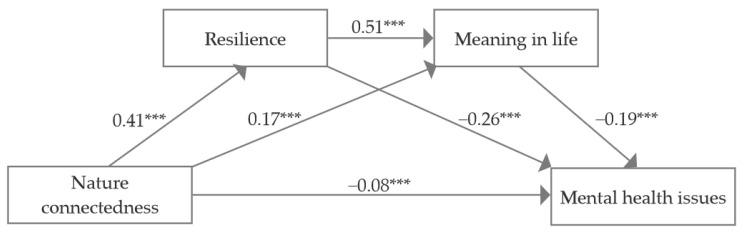
A chain-mediated model of nature connectedness and mental health issues in college students. Note: *** *p* < 0.001.

**Table 1 behavsci-15-00654-t001:** Demographic table of the participants (*n* = 703).

Category		Number (*n*)	Percentage (%)
Gender	Male	375	53.34%
	Female	328	46.66%
Grade	Freshmen	235	33.43%
	Sophomores	243	34.57%
	Juniors	101	14.37%
	Seniors	89	12.66%
	Graduate students	35	4.98%
Home locality	Urban	325	46.23%
	Rural	378	53.77%
Student leader	Yes	306	43.53%
	No	397	56.47%
Family	Only child	262	37.27%
	Have siblings	441	62.73%

**Table 2 behavsci-15-00654-t002:** Summary statistics and correlation analysis of the four variables (*n* = 703).

Variable	*M* ± *SD*	1	2	3	4
1. Nature connectedness	52.98 ± 7.78	1			
2. Resilience	24.72 ± 6.60	0.389 **	1		
3. Meaning in life	48.25 ± 8.92	0.402 **	0.579 **	1	
4. Mental health issues	19.77 ± 13.41	−0.349 **	−0.401 **	−0.434 **	1

Note: ** *p* < 0.01.

**Table 3 behavsci-15-00654-t003:** Regression analysis of nature connectedness and chain mediation model of mental health issues.

Regression Equation		Overall Fit Index			Significance of the Regression Coefficient		
Outcome Variable	Predictor Variable	*R*	*R* ^2^	*F*	*β*	*SE*	*t*
Mental health issues	Nature connectedness	0.50	0.25	37.68 ***	−0.26	0.03	−7.46 ***
Resilience	Nature connectedness	0.40	0.16	22.65 ***	0.41	0.04	11.32 ***
Meaning in life	Nature connectedness	0.63	0.40	65.23 ***	0.17	0.03	5.00 ***
	resilience				0.51	0.03	15.86 ***
Mental health issues	Nature connectedness	0.61	0.38	52.18 ***	−0.08	0.04	−2.23 *
	Resilience				−0.26	0.04	−6.67 ***
	Meaning in life				−0.19	0.04	−5.11 ***

Note: * *p* < 0.5, *** *p* < 0.001.

**Table 4 behavsci-15-00654-t004:** Mediated effect test.

Route	Effect Value	Standard Error	Boot LLCI	Boot ULCI	Effect Size
Nature connectedness → Resilience → Mental health issues	−0.11	0.02	−0.15	−0.07	42.31%
Nature connectedness → Meaning in life → Mental health issues	−0.03	0.01	−0.06	−0.01	11.54%
Nature connectedness → Resilience → Meaning in life → Mental health issues	−0.04	0.01	−0.06	−0.02	15.38%
Direct effect	−0.08	0.04	−0.15	−0.01	30.77%
Total indirect effect	−0.18	0.02	−0.23	−0.14	69.23%
Total effect	−0.26	0.03	−0.33	−0.19	

## Data Availability

If a reasonable request is made, the corresponding author will provide the data for the present study.

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
