# Peer review of "How Does Nature Connectedness Improve Mental Health in College Students? A Study of Chain Mediating Effects"

_behavsci, 2025, doi:10.3390/bs15050654_

Round 1

Reviewer 1 Report

Comments and Suggestions for Authors

This is an interesting paper on the mediators of the relationship between nature connectedness and mental health. It’s a worthwhile topic and provides novel insights. It is generally well written (in broad, structural terms). I make some comments below.

1) l59 – Linguistic mistake: ‘increasing more studies’

2) Page 2, para2 seems to conflate the benefits of nature connectedness with benefits of being in nature. The authors should make this distinction clear.

3) l84 ‘form’ should be ‘from’.

4) l88 no initial is needed in the citation.

5) l88 ‘the’ should be ‘that’.

6) At this point, I stopped noting language errors (there are several others in the manuscript). The authors go back over the manuscript and catch the remaining spelling and grammar mistakes.

7) Section 1.1 does not explicitly touch on college students’ mental health. It should do, because the tile of that section contains ‘College Student’s Metal Health’. It’d be desirable to either add in this content or change the title of the section.

8) l107 ‘the connected’ should be ‘connection’.

9)  l116 Nature connectedness is generally not regarded as an ‘ability’.

10) Section 1.4 - Explain what chain mediation is.

11) l217 It’s not clear what is meant by a ‘random online questionnaire’.

12) Related to my previous point, the Method would benefit from a Procedure section.

13) l354 The data does not suggest that nature connected reduces negative emotions – that implies that the emotions were higher before nature connectedness was increased. This is a subtle wording issue – an issue that can most easily be solved by deleting this sentence.

14) l476 It would be good to expand on why this future research (on students in a specific region) would be desirable. For example, explain why you think these effects might vary by region.

15) Regarding your fourth limitation, more detail is required on this. Are you suggesting that air pollution, heatwaves, etc. might be mediators? If so, elaborate on how they might be.

Comments on the Quality of English Language

As noted above, there are many minor language errors. In a small number of places, they make the meaning hard to understand.

Author Response

Response to reviewers

Dear editors and reviewers,

Thank you for providing us with an opportunity to improve the quality of our manuscript. We appreciate your insightful comments and suggestions. In this revised version, we have addressed all of the points raised. Changes made during the revision process are highlighted in red.

On the following pages, we have provided detailed point-by-point responses to each of the queries raised by the reviewers.

Reviewer #1

Comments 1: l59 – Linguistic mistake: ‘increasing more studies’

Response 1:

This sentence has been removed due to adjustments in the manuscript's logical flow. We have also carefully proofread the entire text and made necessary grammatical corrections.

Comments 2: Page 2, para 2 seems to conflate the benefits of nature connectedness with benefits of being in nature. The authors should make this distinction clear.

Response 2:

The content of the second section has been revised to clarify the concepts of "nature," "nature contact," and "nature connectedness," along with their associated benefits. These modifications are reflected in the updated second and third paragraphs.

Comments 3: l84 ‘form’ should be ‘from’.

Response 3:

The term has been revised to "from" in Line 112.

Comments 4: l88 no initial is needed in the citation.

Response 4:

We have thoroughly checked the citation format throughout the manuscript and removed all first-name initials from references to maintain consistency.

Comments 5: l88 ‘the’ should be ‘that’.

Response 5:

This sentence has been removed due to adjustments in the manuscript's logical flow. We have also carefully proofread the entire text and made necessary grammatical corrections.

Comments 6: At this point, I stopped noting language errors (there are several others in the manuscript). The authors go back over the manuscript and catch the remaining spelling and grammar mistakes.

Response 6:

We sincerely appreciate the reviewer's suggestion. We have carefully proofread the entire manuscript and made corresponding grammatical corrections.

Comments 7: Section 1.1 does not explicitly touch on college students’ mental health. It should do, because the tile of that section contains ‘College Student’s Metal Health’. It’d be desirable to either add in this content or change the title of the section.

Response 7:

We sincerely appreciate the reviewer's valuable suggestion. Recognizing the relative scarcity of research on nature connectedness in college students, we have added a dedicated paragraph in Section 1.1 (lines 119-126) highlighting the significance of studying this phenomenon specifically among college students.

Comments 8: l107 ‘the connected’ should be ‘connection’.

Response 8:

The term has been revised to "nature connectedness" at Line 153 to align with the adjusted content structure of the manuscript.

Comments 9: l116 Nature connectedness is generally not regarded as an ‘ability’.

Response 9:

In accordance with the structural revisions made to the manuscript, the sentence at Line 161-162 has been modified to: "the stronger the sense of nature connectedness of college students".

Comments 10: Section 1.4 - Explain what chain mediation is.

Response 10:

To illustrate using the variables in our study, both resilience and meaning in life can independently mediate the relationship between nature connectedness and mental health, while also potentially operating as sequential mediators. Specifically, nature connectedness may first enhance resilience, which then strengthens the meaning in life, ultimately improving mental health outcomes. We have added this explanatory content at the beginning of Section 1.4 to clarify the proposed mediation pathways.

Comments 11: l217 It’s not clear what is meant by a ‘random online questionnaire’.

Response 11:

We have supplemented detailed participant selection criteria in the second paragraph of Section 2.1 (Procedure) to enhance methodological transparency.

Comments 12: Related to my previous point, the Method would benefit from a Procedure section.

Response 12:

We have added a new subsection, "2.1 Procedure," in the Methods section to provide a detailed description of the experimental protocol.

Comments 13: l354 The data does not suggest that nature connected reduces negative emotions – that implies that the emotions were higher before nature connectedness was increased. This is a subtle wording issue – an issue that can most easily be solved by deleting this sentence.

Response 13:

We sincerely appreciate the reviewer's suggestion. In response, we have removed the statements regarding nature connectedness reducing negative emotions from Section 4.1 to maintain the precision of our findings.

Comments 14: l476 It would be good to expand on why this future research (on students in a specific region) would be desirable. For example, explain why you think these effects might vary by region.

Response 14:

Considering potential variations in geographical and cultural contexts that may influence both the manifestations and degrees of nature connectedness, we acknowledge that these factors could lead to divergent research outcomes. This limitation has been explicitly addressed in the newly added content (Lines 533-536) as the third point of discussion in our study's limitations section.

Comments 15: Regarding your fourth limitation, more detail is required on this. Are you suggesting that air pollution, heatwaves, etc. might be mediators? If so, elaborate on how they might be.

Response 15:

We sincerely appreciate the reviewer's insightful comment. We would like to clarify that these objective factors were not intended to serve as mediator variables in our study. The confusion stemmed from our initial insufficient elaboration on the operational definition of "nature" in the manuscript. To address this, we have:

  1. Provided a clearer conceptual definition of "nature" in the second paragraph;
  2. Removed the potentially misleading content to prevent reader misinterpretation.

We have made every effort to address the reviewers' comments and improve the manuscript accordingly. We sincerely appreciate the Editors and Reviewers for their time and constructive suggestions. We hope that our revisions have satisfactorily responded to all the concerns raised.

Thank you once again for your valuable feedback.

Sincerely,
Mei Zhao
zhaomei@psych.ac.cn
(On behalf of all the authors)

Reviewer 2 Report

Comments and Suggestions for Authors

Thank you very much for this interesting approach to the topic of "nature connectedness and mental health".

Please take into account the following recommendations:

1.) In many places in the text there are no references, or where there are, only one reference is given (e.g. in the sentences starting in row 57, 60, 73,... but throuout the whole text). In my opinion, this is related to an incomplete discussion of the scientific-theoretical foundations on the topics of ‘Definition of nature’, ‘Definition of connectedness to nature’, “Resilience”, ‘Stress reduction’, etc. Individual theories are mentioned for many aspects of relationships and humanity, but these are usually only referenced with one source. It is not argued why this one theory was chosen (and not another) and the pros and cons of the respective theories are not weighed up. Here is an example: The ‘attention restoration theory’ is listed, but only mentioned with one source, not described in detail, not explained why this particular theory was chosen and no other, and what speaks for us against this theory. To summarise, the choice of sources and scientific theoretical foundations was very selective and non-transparent. This leads to generalised statements and the end of each chapter about the connections between nature, closeness to nature, resilience and mental health, which are not scientifically comprehensible based on the sources and theories cited.

2.) In my opinion, some statements are also simply wrong, e.g. in lines 73-74: I don't automatically focus on natural elements just because I'm in nature. Such simplistic and unscientific statements should be avoided. Or in row 137: Philosophers and theologians have been exploring the meaning of life for thousands of years. You can't attribute that to one person.

3.) Some sentences I don't understand, or it doesn't contain errors or phrases that indicate that a translation tool was used. I recommend having the text proofread by a native speaker.

4.) Chapter 1.5: Hypotheses are named, but no reserach questions.

5.) Chapter methods, row 214: I do not understand this sentence. The methods are not clear to me.

6.) In my opinion, the team of authors does not distinguish clearly enough throughout the paper between a subjective connection to nature on an emotional level and actual time spent in nature (frequency, duration, etc.). In my opinion, these are two different pairs of shoes that need to be clearly separated: I can spend a lot of time in nature but have no connection to it. Or I can rarely be in nature but feel connected to it. The theories they put forward also refer to one and then the other. There needs to be a clearer differentiation here.

Comments on the Quality of English Language

see above

Author Response

Response to reviewers

Dear editors and reviewers,

Thank you for providing us with an opportunity to improve the quality of our manuscript. We appreciate your insightful comments and suggestions. In this revised version, we have addressed all of the points raised. Changes made during the revision process are highlighted in red.

On the following pages, we have provided detailed point-by-point responses to each of the queries raised by the reviewers.

Reviewer #2

Comments 1: In many places in the text there are no references, or where there are, only one reference is given (e.g. in the sentences starting in row 57, 60, 73,... but throuout the whole text). In my opinion, this is related to an incomplete discussion of the scientific-theoretical foundations on the topics of ‘Definition of nature’, ‘Definition of connectedness to nature’, “Resilience”, ‘Stress reduction’, etc. Individual theories are mentioned for many aspects of relationships and humanity, but these are usually only referenced with one source. It is not argued why this one theory was chosen (and not another) and the pros and cons of the respective theories are not weighed up. Here is an example: The ‘attention restoration theory’ is listed, but only mentioned with one source, not described in detail, not explained why this particular theory was chosen and no other, and what speaks for us against this theory. To summarise, the choice of sources and scientific theoretical foundations was very selective and non-transparent. This leads to generalised statements and the end of each chapter about the connections between nature, closeness to nature, resilience and mental health, which are not scientifically comprehensible based on the sources and theories cited.

Response 1:

We sincerely appreciate the reviewer's professional comments. In response to your suggestions, we have made the following revisions to the manuscript:

  • Additional references related to variables have been included, such as references supporting nature, nature exposure, nature connectedness, psychological resilience, and sense of meaning in life.
  • Additional references related to theories have been added, such as the Biophilia Hypothesis (in the third paragraph).
  • The wording at the end of the literature review sections has been revised, such as in sections 1.2, 1.3, and 1.4.

Please refer to the red-colored text in the manuscript for the specific adjustments made.

Comments 2: In my opinion, some statements are also simply wrong, e.g. in lines 73-74: I don't automatically focus on natural elements just because I'm in nature. Such simplistic and unscientific statements should be avoided. Or in row 137: Philosophers and theologians have been exploring the meaning of life for thousands of years. You can't attribute that to one person.

Response 2:

We sincerely appreciate the reviewer's insightful comment. In response, we have removed the problematic statement originally located in Lines 73-74 of the manuscript. We have revised the original statement in Line 137 and further elaborated on this point in Lines 180-181 to provide clearer theoretical justification.

Comments 3: Some sentences I don't understand, or it doesn't contain errors or phrases that indicate that a translation tool was used. I recommend having the text proofread by a native speaker.

Response 3:

The manuscript has undergone thorough proofreading by a native English-speaking scholar, with subsequent revisions made throughout the text to ensure linguistic accuracy and academic rigor.

Comments 4: Chapter 1.5: Hypotheses are named, but no research questions.

Response 4:

We have incorporated a clear articulation of the research questions in the fourth paragraph to better frame our research questions.

Comments 5: Chapter methods, row 214: I do not understand this sentence. The methods are not clear to me.

Response 5:

The statements related to the sample size calculation method have been removed from the text.

Comments 6: In my opinion, the team of authors does not distinguish clearly enough throughout the paper between a subjective connection to nature on an emotional level and actual time spent in nature (frequency, duration, etc.). In my opinion, these are two different pairs of shoes that need to be clearly separated: I can spend a lot of time in nature but have no connection to it. Or I can rarely be in nature but feel connected to it. The theories they put forward also refer to one and then the other. There needs to be a clearer differentiation here.

Response 6:

While this study focuses on the overall impact of nature connectedness on mental health without distinguishing between short-term and long-term effects, we recognize this as a valuable direction for future research. This limitation has been explicitly addressed as the fourth point in the study's limitations section (Lines 536-539).

We have made every effort to address the reviewers' comments and improve the manuscript accordingly. We sincerely appreciate the Editors and Reviewers for their time and constructive suggestions. We hope that our revisions have satisfactorily responded to all the concerns raised.

Thank you once again for your valuable feedback.

Sincerely,
Mei Zhao
zhaomei@psych.ac.cn
(On behalf of all the authors)

Round 2

Reviewer 2 Report

Comments and Suggestions for Authors

Thank you very much for revising the manuscript so rapidely. In my eyes, it has gained a lot of strength.

However, I still miss reserach questions in Chapter 1.5 and can only find hypotheses there.

Comments on the Quality of English Language

While reading the manuscript, I still have the impression that the style of writing is sometimes rather "narrative", using unscientific formulations and phrases/words, that I would rather expect in a story than in a scientific manuscript.

Author Response

Response to reviewers

Dear editors and reviewers,

Thank you for providing us with an opportunity to improve the quality of our manuscript. We appreciate your insightful comments and suggestions. In this revised version, we have addressed all of the points raised. Changes made during the revision process are highlighted in red.

On the following pages, we have provided detailed point-by-point responses to each of the queries raised by the reviewers.

Reviewer

Comments and Suggestions for Authors:

Thank you very much for revising the manuscript so rapidely. In my eyes, it has gained a lot of strength. However, I still miss reserach questions in Chapter 1.5 and can only find hypotheses there.

Response:

Thank you for the reviewer's suggestion. I have added the research question in Section 1.5, lines 244-248.

Comments on the Quality of English Language:

While reading the manuscript, I still have the impression that the style of writing is sometimes rather "narrative", using unscientific formulations and phrases/words, that I would rather expect in a story than in a scientific manuscript.

Response:

The entire manuscript has been revised according to the suggestion, with efforts made to align it with the language style of a scientific manuscript.

Sincerely,
Mei Zhao
zhaomei@psych.ac.cn
(On behalf of all the authors)